# Transfer Incremental Learning using Data Augmentation

**Ghouthi Boukli Hacene, Vincent Gripon, Nicolas Farrugia, Matthieu Arzel & Michel Jezequel**
Department of Electronics
IMT Atlantique
`{name.surname}@imt-atlantique.fr`

## Abstract

Due to catastrophic forgetting, deep learning remains highly inappropriate when facing incremental learning of new classes and examples over time. In this contribution, we introduce Transfer Incremental Learning using Data Augmentation (TILDA). TILDA combines transfer learning from a pre-trained Deep Neural Network (DNN) as feature extractor, a Nearest Class Mean (NCM) inspired classifier and majority vote using data augmentation on both training and test vectors. The obtained methodology allows learning new examples or classes on the fly with very limited computational and memory footprints. We perform experiments on challenging vision datasets and obtain performance significantly better than existing incremental counterparts.

## 1 Introduction

Humans have the ability to incrementally learn new pieces of information through time, building over previously acquired knowledge. This process is most of the time nondestructive, and results in what is often referred to as "curriculum learning" in the literature (Bengio et al. (2009)). On the contrary, it has been known for decades that neural networks learning procedures, despite the fact they originally were proposed as a simplifying model for brain mechanisms, suffer from "catastrophic forgetting" (Kasabov (2013); French (1999)), or the fact that previously learned knowledge is destroyed when learning new one.

Incremental methods provide solutions to process data sequentially. We define incremental learning approaches by stating they should have (Rebuffi et al. (2017)): a) an ability to learn new data providing additional information (example-incremental) or new classes (class-incremental), b) an ability to reach a classification accuracy comparable to state-of-art nonincremental counterparts, c) an ability to preserve previous knowledge without needing to access previously seen data. Most of existing works either add new classifiers to accommodate new data, such as the learn++ method (Polikar et al. (2001); Sun et al. (2016)), or retrain the model using newly received data together with the old model (Syed et al. (1999); Poggio & Cauwenberghs (2001)). Other methods end up with a high memory footprint, such as Budget Restricted Incremental Learning (BRIL) (Hacene et al. (2017)), and/or a poor classification accuracy compared to state-of-art, such as Incremental Classifier and Representation Learning (iCaRL) (Rebuffi et al. (2017)) or Nearest Class Mean (NCM) classifiers ( Mensink et al. (2013)).

In this paper we propose an incremental learning method with the following claims: a) it performs incremental learning as previously defined, b) it requires small computational power and memory footprints compared to existing counterparts, and c) it approaches state-of-art accuracy on challenging vision datasets (CIFAR10, CIFAR100 and ImageNet). It builds upon already proposed methods, including transfer learning Girshick et al. (2014); Pan & Yang (2010), NCM classifiers and also make use of data augmentation at both training and testing phases.

## 2 Method

TILDA is made of four main ingredients: 1) a pre-trained DNN to perform feature extraction of signals, 2) a vector splitting to project features into low dimensional subspaces, 3) an assembly of

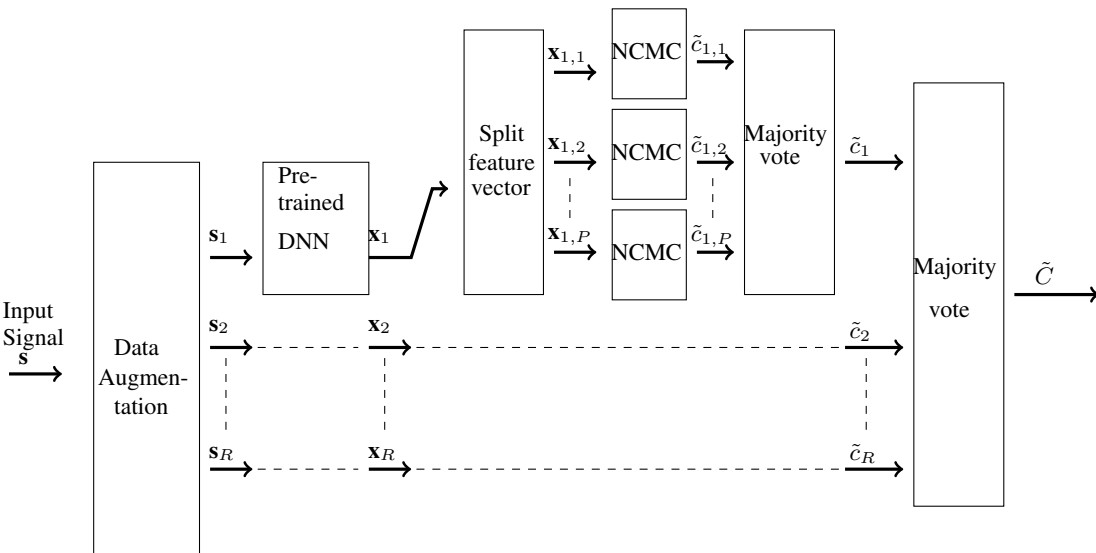

Figure 1: *Overview of TILDA during prediction. The input signal $s$ is first derived in multiple versions $(s_r)_{1 \leq r \leq R}$ using data-augmentation. Then, a pre-trained DNN is used on each version to extract corresponding features $(x_r)_{1 \leq r \leq R}$. Subsequently, we split each feature vector $x_r$ into P parts $(x_{r,p})_{1 \leq p \leq P}$ of roughly the same dimension. Each one is classified using an NCM-based technique to obtain a weak decision $(\tilde{c}_{r,p})_{1 \leq p \leq P}$. Finally, a majority vote is processed aggregating the decision for each part then a second one for each version of the input signal.*

NCM-inspired classifiers (NCMC) applied independently in each subspace and 4) a data augmentation inspired scheme to increase accuracy of the classifying process. These steps are summarized in Figure 1 and detailed in the next paragraphs.

The first step is to use the internal layers of a pre-trained DNN as a generic feature extractor and compute the feature vector $\mathbf{x}^m$ corresponding to the input signal $\mathbf{s}^m$. This is often referred to as "transfer learning" in the literature ( Girshick et al. (2014); Pan & Yang (2010)). Then each feature vector $\mathbf{x}^m$ is split into $P$ subvectors of equal size denoted $\left(\mathbf{x}_p^m\right)_{1 \leq p \leq P}$.

During training, for each class $c$ and each subspace $p$, we produce $k$ anchor vectors $Y_{c,p} = [\mathbf{y}_{c,p,1}, ..., \mathbf{y}_{c,p,k}]$, and their associated counters $N_{c,p} = [n_{c,p,1}, \ldots, n_{c,p,k}]$. Algorithm 1 summarizes the learning process.

---

**Algorithm 1** Incremental Learning of Anchor Subvectors

---

**Input**: streaming feature vector $\mathbf{x}_c^m$
    **for** $p := 1$ to $P$ **do**
        Randomly select $i$, where $1 \leq i \leq k$
        $\mathbf{y}_{c,p,i} \leftarrow \mathbf{y}_{c,p,i} + \mathbf{x}_{c,p}^m, n_{c,p,i} \leftarrow n_{c,p,i} + 1$
    **end for**

---

During prediction phase, in each subspace anchor vectors are divided by their corresponding counters and are used for nearest-neighbor search. More precisely, each subpart $\mathbf{x}_p$ obtained from input signal $\mathbf{s}$ is weakly classified using the nearest-neighbor anchor vector in part $p$. We finally perform a majority vote to aggregate these weak classifications.

To further improve the method performance, we use data augmentation at both prediction and training phases. The training data-augmentation is used to artificially enrich the training dataset, whereas the prediction data-augmentation is used to obtain multiple decisions that are aggregated using a second majority vote.

**Remark**: when facing class-incremental scenarios (all examples of a class are given at the same time, but new classes occur over time), we rather use $k$-means instead of Algorithm 1 to specify anchor vectors.

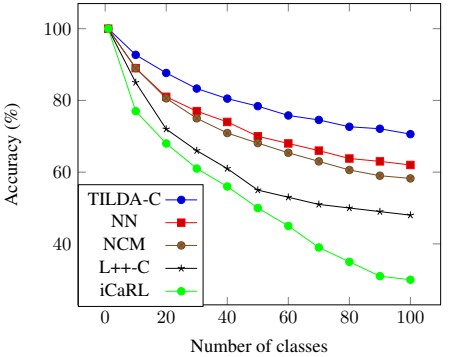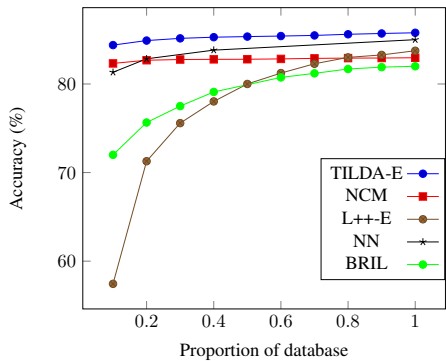

*Figure 2: Evolution of the accuracy as a function of number of classes (CIFAR100) (left) and as a function of the number of learning examples (CIFAR10) (right).*

## 3 EXPERIMENTS

We compare our method to BRIL (Hacene et al. (2017)), NCM (Mensink et al. (2013)), iCaRL with ResNet32 as described in Rebuffi et al. (2017), Nearest Neighbour search (NN), and learn++ (Polikar et al. (2001)) using Classification And Regression Trees (CART) weak classifiers, in class/example incremental learning scenario. We also compare our method to non-incremental techniques (NI): transfer learning with a multilayer perceptron (TMLP) and a fully trained ResNet32 on raw data (R32). For all incremental methods but iCaRL, Inception V3 is used as pre-trained DNN. Table 1 shows the obtained results. Figure 2 depicts the evolution of accuracy of several methods as a function of time, when classes are added one at a time (Class-Incremental (CI)) or when examples from all classes are added 5000 at a time (Example-Incremental (EI)). TILDA uses $P = 16$ and $k = 30$. We add a $C$ (resp. an $E$) after a given method (e.g. TILDA-C) to refer to the class-incremental (resp. example-incremental) version of this method. Results show that TILDA provides a good trade-off between accuracy and memory footprint, being almost on par with non-incremental counterparts.

*Table 1: Comparison of accuracy (Acc) and memory usage (M) relative to full dataset (corresponding to* $100\%$*) for the different methods. Note that memory usage of Learn++ method represents the size of weak classifiers. Learn++ (L++) adds* $5/1$ *weak classifiers for CIFAR10/100 at a time. First two rows (resp. last two) correspond to results on CIFAR100 (resp. CIFAR10).*

|  | only CI | | | both CI and EI | | | | only EI | NI | |
|---|---|---|---|---|---|---|---|---|---|---|
|  | TILDA-C | L++-C | iCaRL | NN | NCM | BRIL | TILDA-E | L++-E | TMLP | R32 |
| Acc % | **70.5** | 48 | 30 | 60.2 | 58.25 | 57 | **63.4** | 42 | 68.6 | 68.6 |
| M % | **6** | 16 | **6** | 100 | **0.2** | 6 | 6 | 22 | 100 | 100 |
| Acc % | **89.2** | 79 | 29 | 85 | 83 | 82 | **85.8** | 83.7 | 90 | 91 |
| M % | **0.6** | 1.6 | **0.6** | 100 | **0.02** | 0.6 | 0.6 | 0.6 | 100 | 100 |

The obtained 5-top accuracy on ImageNet ILSVRC 2012 by TILDA with $k = 30$ is **93/92%** on class/example incremental learning scenarios, which approaches state-of-art accuracy.

## 4 CONCLUSION

We introduced a novel incremental learning method based on data augmentation, pre-trained DNNs, weak classifiers based on NCM or $k$-means, and two majority votes. This combination of methods allows to learn incrementally new example/classes using very low memory usage. The results show that TILDA outperforms other incremental methods on CIFAR-10/100 datasets, even being on par with non-incremental methods. As a consequence, we believe this method is promising for embedded devices. Future work includes proposing efficient hardware implementations of it.

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
