# OpenReview forum: "Transfer Incremental Learning using Data Augmentation"
_ICLR.cc/2018/Workshop — Reject_

### Official Review · AnonReviewer2 · 2018-03-02
**Using NCM style classifiers on subvectors for incremental learning, yet unclear if results are just due to pre-trained DNN.**

**Rating:** 4
**Confidence:** 5

**Review:**

This paper proposes to use a NCM style algorithm for class/example incremental learning.
It starts from a pre-trained ConvNet, then split feature vector into P subvectors, use NCMC (NCM with multiple class representatives) with k-centroids, and then a majority vote over the P subverters for the class predictions. To improve results at both train and test-time data augmentation is used.

I have two major problems with the current submission:
1) Due to the use of the pre-trained DNN comparison to (eg) iCARL are unfair, which explicitly aims to train the DNN while learning classifiers. Moreover, the results on ImageNet are skewed as well, given that the DNN is also pre-trained on ImageNet. In conclusion: using pre-trained DNN makes comparison to related methods unclear/unfair, and makes some experimental results of little added value.

2) The important aspects of the proposed method are not explored: influence of data augmentation, of number of subvectors  P, of number of NCMC centroids k. Moreover, it is unclear what is "learned" during training. From the setup it seems that just the means of classes (or k-means centroids) are obtained. Why does that work well with random assignment (Algorithm 1)?

Minor issues:
- Unclear how it relates to iCARL/NCM etc, does the method reduces to (say) iCARL when data augmentation is not used, P=1, and K=1.
- Unclear what the baselines are exactly, what is learned with NCM? How well is the NCMC baseline? Does NCM(C) improve with data augmentation? Or with subvectors?
- Experimentally unclear if the performance increase is just due to the DNN (ie which methods are based on the pre-trained DNN?

---

### Official Review · AnonReviewer1 · 2018-03-10
**no title**

**Rating:** 6
**Confidence:** 2

**Review:**

[ Paper Summary ]

The paper proposed NCM based classifier for incremental learning, which combines pre-trained DNN, data augmentation, feature splitting, and NCM.

- novelty

The approach seems to be novel, though the topic is not my expertise.

- clarity

The procedure is clear, though it is a bit difficult for those who are not familiar with NCM.

- significance

The problem setting would be significant, and performance on experiments is good.

- quality

The entire quality of the paper would be ok, and the main claims are supported by the results.

[ Comments ]

- pros

The method works well in the experiments.

The method seems to be easy to implement.

- cons

The effect of each of four components in the proposed method is not clear. Discussing component-wise effect might be informative.

Robustness to P would be important.

---

### Official Review · AnonReviewer3 · 2018-03-11
**insufficient technical contribution**

**Rating:** 3
**Confidence:** 3

**Review:**

This paper proposes a method for incremental learning of deep neural networks while adding new samples and new classes.  To avoid drift and forgetting, a set of anchor points are maintained via a nearest neighbours approach.

+ somewhat compelling results on CIFAR10/100 and ImageNet
- insufficient technical novelty and contribution

---

### Decision · Program_Chairs · 2018-03-20
**ICLR 2018 Workshop Acceptance Decision**

**Decision:**

Reject

**Comment:**

Based on the reviews, this paper has not been accepted for presentation at the ICLR workshop. However, the conversation and updates can continue to appear here on OpenReview.